# Towards Dynamic Model-Based Agile Architecting of Cyber-Physical Systems

**DOI:** 10.3390/s22083078

**Published:** 2022-04-17

**Authors:** Alexander Vodyaho, Nataly Zhukova, Alexey Subbotin, Fahem Anaam

**Affiliations:** 1Department of Computer Science and Engineering, Saint-Petersburg State Electrotechnical University, 197376 St. Petersburg, Russia; aivodyaho@mail.ru (A.V.); fahemye@gmail.com (F.A.); 2St. Petersburg Federal Research Center of the Russian Academy of Sciences (SPC RAS), 199178 St. Petersburg, Russia; nazhukova@mail.ru

**Keywords:** digital dynamic threads, digital twins, IoT systems architecting, model synthesis, multi-level relatively finite state automaton

## Abstract

A model-based approach to large-scale distributed system architecting is suggested, which is based on the use of dynamic digital twins. This approach can be considered as an integration of known paradigms, such as digital twins, evolutionary architecture and agile architecture. It can also be considered as one of the possible realizations of the digital thread paradigm. As part of this approach, a three-level digital thread reference architecture is suggested, which includes the following levels: (i) digital thread support level; (ii) agile architecture support level; (iii) digital shadow support level. This approach has been used in the development of a number of real systems, and has shown its effectiveness in supporting system agility at the exploitation and modernization stages. The proposed approach is focused on building digital twin-based systems. This article may be interesting for specialists engaged in research and development in the domain of IoT- and IIoT-based information systems, primarily architects.

## 1. Introduction

The main driving forces behind the current stage of the development of technology, along with nanotechnology, are advances in microelectronics, telecommunications and software engineering. At the same time, the complexity of the man-made systems created is constantly increasing, which is expressed not only, and not so much, in an increase in the number of elements, and the number of internal and external links, as in a constant increase in the level of their variability. Another important feature of modern man-made systems is that they are built from elements of a different physical nature. These are cyber-physical systems (CPS), socio-cybernetic systems, natural systems, etc. [1,2,3,4,5]. 

Modern man-made systems implement a complex set of functions, many of which can be classified as intelligent. These functions can only be implemented through the presence of a powerful software component in the system.

Many modern systems can be classified as information-oriented systems. The development of microelectronics makes it possible to use many sensors and processors as part of systems, which leads to the need to accumulate and process large amounts of data. This is a well-known big data problem [6]. 

Based on the above, it can be argued that the distinctive features of modern systems are high complexity, a high level of structural and functional variability, as well as a high level of heterogeneity. Thus, a modern system can be defined as a large-scale complex distributed system (LSCDS), and can be considered as a system of systems (SOS) [7]. LSCDS are information-oriented systems. From the point of view of the implemented set of functions, they are also intelligent systems. Designing such systems requires the use of new approaches and paradigms, which take into account the high level of structural and functional dynamics. This can be performed with the help of a model-based approach.

This paper considers one of the possible approaches to the development of distributed systems built on IoT platforms. The paper has two research objectives (RO): RO1 is the development of approaches to implement mechanisms of digital threads (DTh), and RO2 is the development of mechanisms to support run time (RT) agility.

The article includes 11 sections. Section 2 discusses the main paradigms of LSCDS development currently used. Section 3 describes the DTh approach. In Section 4, the problems of DTh-based LSCDS architecting are considered, and the main tasks that need to be solved for the implementation of the DTh approach are discussed. Section 5 describes the proposed dynamic DTh approach. Section 6 provides formal definitions of the key terms of the suggested approach, such as dynamic DTh and dynamic DT. Section 7 describes the DTh evolutionary reference architecture. Section 8 gives a description of the generalized algorithm of multilevel monitored and managed system (MMS) model synthesis. Section 9 estimates the degree of readiness of the proposed approach for practical use. Section 10 provides an example of the use of the proposed approach. The final part (Section 11) contains conclusions and a description of possible directions for further research.

## 2. Modern Paradigms of LSCDS Development

Newly created systems, the number of which is increasing, can be classified as large and complex systems, as well as SoS [7]. An increase in the complexity of systems leads to an increase in the cost and time spent developing such systems. This, in turn, leads to the need to increase their lifetime, with the aim of increasing the ROI, which is achieved through their constant modernization. It should be noted that the ability to modernize should be laid down at the stage of creating the system. This approach is known as evolutionary architecture (EA) [8]. 

Currently, a relatively large number of paradigms and approaches to solving problems related to the complexity and variability of LSCDS are known. These include the following: model-driven architecture (MDA) [9], EA [8], agile architecture (AA) [10,11], DevOps [12], DT [13], DTh [14,15], and ambient intelligence (AmI) [4]. 

The above paradigms relate to different phases of the life cycle (LC), but they all make use of a model-based approach somehow. 

One of the key ideas of LSCDS development is the idea of DTh, which is essentially an integration approach that can be considered as an umbrella technology for modern paradigms of LSCDS development (Figure 1).

## 3. Digital Threads

The DTh concept is a rather complex and multidimensional concept. There is a fair few different definition of this concept. According to the most general definition, “the digital thread is an integrated view of everything about an asset or product throughout its lifecycle that enables improved communication and collaboration” [14]. Quite often, the definition of a DTh relates to the production phase, e.g., according to [15], a DTh is “a communication framework that connects traditionally siloed elements in manufacturing processes and provides an integrated view of an asset throughout the manufacturing lifecycle”. Taking into account the fact that the idea of a DTh is closely related to the idea of the digital transformation of society, the first definition seems more useful from the point of view of developing approaches for practical implementation. This allows us to consider the DTh concept as one of the possible approaches to realizing digital transformations.

## 4. Problems of DTh-Based LSCDS Architecting

Architecting is an important stage of LSCDS development [16]. Wrong solutions implemented at this stage are quite difficult to correct. The problem is that when using DTh-based approaches to build LSCDS, in addition to developing and maintaining the system itself, it is necessary to develop and maintain its models. If the models for each stage of LC are developed independently, this can lead to additional expenses. Thus, the task of model transformation comes to the forefront, but the models related to different stages of the LC differ significantly. In this context, automatic construction and the transformation and maintenance of up-to-date models related to different phases of the LC system pose a problem.

## 5. Proposed Approach

The idea of the proposed approach is the joint use of three paradigms: DTh, EA and AA. The use of DT can reduce the total cost of ownership. EA is considered a means of adapting the system to changes in the external environment. AA is considered a tool for maintaining run time agility. The proposed approach can be defined as an agile or dynamic digital threads (DDTh) approach. 

The approach is based on a number of known paradigms and approaches, such as the following: (i) a model-based approach, (ii) continuous architecture, and (iii) agile architecture, and can be considered as a fusion of these paradigms.

The suggested DDTh paradigm can be considered as the realization of a continuous architecture paradigm [17], and is based on the dynamic DT paradigm developed by the authors [18], which can be considered as a further development of this approach.

## 6. Dynamic DTh and Dynamic DT

DTh can be defined as *DTh = <DTE, Tr>,* where *DTE* is a set of digital elements and *Tr* is a set of transformation procedures. *DTE* describes elements related to a certain model level. *DTE* can be defined as *DTE = <E, DT, IE, EI>,* where *E* is some entity of any nature, *DT* is a digital twin, *IE* is an internal interface (*E* − *DT*), and *EI* is an external interface (*DT* —the external world). 

The main *DTE* types that correspond to the LC phases are the following: *LCP = <DTDE, PDTE, EDTE, MDTE>,* where DTDE is a DTE which is related to the development phase, PDTE is a DTE which is related to the production phase, EDTE is a DTE which is related to the operation phase, and MDTE is a DTE which is related to the modernization (upgrading) phase.

*DT* is the model of E. Entities can have the following different natures: virtual entity (V), physical entity (P), biological entity (B), social entity (S), etc. Table 1 shows the main types of models.

The key concept for the DDTh is the concept of the DDT, which is a multidimensional concept. 

Depending on the point of view, there are many definitions of DT [13]. 

In general, a DT is defined as “a virtual representation of an object of any nature which is linked in a bidirectional way with the counterpart”. A digital twin can be considered as a set of models to support all stakeholders during the whole lifecycle of the object it represents. 

It is not only a physical object, but a process or system that consists of different natural elements, such as humans, that can be represented with the help of DT. 

The state of the MMS related to a certain point in time in this article is defined by the term digital shadow (DS). 

The following points are important in defining DT: (i) different models are used at different stages of the LC, which are built to solve different tasks; (ii) DT can be built in terms of data, information or knowledge; and (iii) DT can be built using different methodologies. 

Depending on the LC phase, one can employ different definitions of DT, such as the following: (i) DT is a model (a system of models) describing the E at a certain stage of the LC; (ii) DT is a port of access to other systems and worlds, in particular, physical ones; (iii) DT is an intelligent controller; (iv) DT is an architectural style; and (v) DT is a user interface adapter. 

In the first case, when we are talking about the implementation of the concept of a DTh, DT is considered as a means of implementing the DTh mechanisms. 

In the second case, DT is considered as a means of constructing SoS. Each system that is part of SoS can have an arbitrary number of DTs that describe the system in a language that is understandable to other systems. In this case, DT can be considered as a service available to other systems that are part of SoS. 

In the third case, DT is considered as a means of implementing the agility mechanism. DT manages the MMS structure and behavior reconfiguration. 

In the fourth case, DT is considered from the point of view of software architecture. DT can be viewed as one of the possible implementations of the virtual machine’s architectural style [19]. 

In the fifth case, the system consisting of MMS and DT is considered a single system, which is accessed through DT. This is the user’s point of view. 

These points of view can be considered as architectural points of view [16].

## 7. DDTh Reference Architecture

EA can be defined as EA = <EAp>, where EAp is a set of architectural points of view. Obviously, each point of view is described by its own set of models. In this case, the architectural process can be described as a relatively finite state operating automaton (RFSOA). 

As a formal model describing the DDTh architectural design process, it is proposed that a three-level model should be used, which includes the following levels (Figure 2): (i) DDTh support level; (ii) AA support level; and (iii) the level of working with DS. 

The upper level (DDTh support level) describes the architectural design process. A probabilistic RFSOA is used for this purpose.

The RFSOA is an automaton with a variable structure, in which the functions of transitions and outputs explicitly depend on time [20]. Each state of the automaton corresponds to a certain architectural state Si = <MMSi, MMMSi>, where MMSi is the state of MMS and MMMSi is the model corresponding to this state. Each state corresponds to a phase of the LC. At this level, the system’s modernization abilities are described by a probabilistic RFSOA. The probabilities correspond to the probabilities that there will be a need to switch to a specific architectural state during the modernization process. The architectural states are defined by the architect, and it is assumed that they are known.

At the intermediate level (AA support level), the RT reconfiguration processes are realized. At this level, an RFSOA can also be used. In this case, the RFSOA states describe the architectural states, which are a high-level structural and functional description of the MMS. Fitness functions [8] can be used to describe transitions between architectural states. 

At the lower level, models that are focused on working with DS are used, i.e., determining the current state of the MMS. For this purpose, models describing the structure of MMS and business processes occurring in it are used. At the lower level, data acquisition procedures are implemented. The collected data can be presented in the form of models. This mechanism is discussed in sufficient detail in [20,21,22,23,24].

## 8. Generalized Algorithm of Multilevel MMS Model Synthesis

The proposed approach assumes the use of multilevel dynamic models. This applies to all three levels of the DDTh model. The RFSOA acts as a model. Working with dynamic models requires the use of mechanisms for constructing models (synthesis) based on events. 

To solve the problems associated with multilevel synthesis, a generalized algorithm is proposed that allows object models to be synthesized in accordance with the set of synthesis goals and specified efficiency criteria. This algorithm can be used to investigate the object, in order to restore its model, i.e., determine all the parameters of the RFSOA describing the MMS. A detailed description of the algorithm is given in [20,23]. The general structure of the algorithm is shown in Figure 3. Along with well-known process mining algorithms, it is one of the basic algorithms of DT model synthesis.

It is assumed that the input information for the synthesis procedure comes in the form of event streams (logs). The task of clearing different kinds of noises from the log is realized at the preprocessing stage. Sometimes, such clearing is not a trivial task, e.g., when working with cascading failures [23]. Different approaches can be used for preprocessing the event streams, e.g., one can use rule-based systems or neural networks. 

According to the algorithm, the first step is to establish an MMS model synthesis problem statement. The second step is model synthesis (reconstruction). The third step is the evaluation of the synthesized model. If the model quality does not meet the requirements, we realize the procedure of data acquisition processes synthesis (restructuring) and return to the second step; otherwise, we go to the synthesis of MMS model representations. This last step in the synthesis of the MMS model is building the necessary form of the model representations, e.g., a knowledge graph.

During processing, the characteristics of the content of the streams are determined, and their features are revealed. The processing context is determined by the state of the environment in which the objects function, the processing conditions, and the capabilities of the data collection systems used. The available a priori information about the objects and accumulated statistical data can play a significant role in processing. 

Based on the results of data processing, new MMS models are built or existing models are rebuilt. In a highly dynamic environment, where objects exhibit complex behavior, the restructuring of object models is performed quite often. For the constructed models, their parameters are determined and the models are evaluated. Inconsistencies are revealed between the models reflecting the actual state of objects, which are determined on the basis of information extracted from the content, and models necessary for solving practical problems. If the synthesized models do not comply with the required models, then the data necessary for the reconstruction of existing models are collected. In this case, monitoring processes and corresponding monitoring programs are to be synthesized. A cycle involving the reconstruction of models of objects, processes and programs is performed until the required models are built or it is proved that their construction is impossible. The task of MMS synthesis is considered to be solved if the resulting models provide a solution to the applied problem and the obtained solutions satisfy the efficiency criteria.

## 9. The Degree of Readiness of the Proposed Approach for Practical Use

The degree of readiness can be estimated as an average. Currently, the following elements have been developed: (i) general concept of DDTh; (ii) model of RT agility support, in the form of RFSOA and algorithms for its synthesis; (iii) lower-level special case models for descriptions of structural and behavioral variability [25]. At present, the DDTh approach described can be used at the exploitation and modernization stages.

## 10. Case Study

The approach described above was used to solve a number of real-life problems in the field of cyber-physical systems, such as in production systems [26,27], a cable television network MMS [28], and a subway information flow management system [29]. The latter of these use cases is described below. This use case shows the possibility of using the proposed approach at different stages of the MMS LC, namely, at the stages of operation and modernization. In addition, this approach is interesting from the point of view of solving the problem of transition from traditional systems to systems using DT. In this case, the DT model is presented in the form of a knowledge graph, since the developed models should be a fragment of the knowledge graph.

This project is part of a big project entitled “Smart City”, one of the goals of which is the management of transport and passenger flows in St. Petersburg (Russia). Within the framework of this project, the task of developing a smart system for monitoring the state of the subway infrastructure was solved.

The stakeholders included the technical support specialists, responsible for the technical condition of the subway infrastructure, as well as the architects of the smart transport system, who were primarily interested in monitoring and analyzing data related to passenger traffic in the subway. 

It was obvious that a single MMS should be used to solve these problems. When creating a subway infrastructure management system, two main problems arose. The first was organizing a smooth transition from the existing infrastructure to a smart infrastructure, while being required to gradually update the sensors and actuators used.

The second problem was that when work on the project began, the architects of the smart transport system could not formulate the requirements for the volume and formats of the necessary data specifically enough; however, it was pointed out that the smart transport system is focused on the active use of knowledge management mechanisms.

Taking into account the requirements listed above and uncertainties, it was decided to use the most flexible solutions based on the model approach. It was supposed that DF could be used to solve the problem of having to permanently upgrade both the equipment and software. Moreover, it was supposed that the use of an agile architectural approach, in which models are implemented in terms of knowledge, would simplify the task of integration with other subsystems that are part of the smart transport ecosystem.

The proposed approach was used for data stream processing in the subway’s intelligent video surveillance system. In the subway, sensors are used together with video cameras, which are placed on objects (Figure 4). The object can be an escalator, a platform, a tunnel, or the ground area of a subway station around the entrance to the subway or depot. Information via Wi-Fi routers (twisted pair, optics, etc.) enters the cloud or fog (servers, deployed in cloud or fog), where it is processed and stored, models are built, and information is then visualized on a PC or smartphone. The whole system has a flexible structure; when one object fails, the whole system does not stop working. For information, visualization programs for different operating systems on PCs (Windows, Linux, and macOS) and smartphones (Android and iOS) have been developed.

The video of the movement of people and the tape of the subway escalator were taken from open sources (YouTube, Yandex.Video, etc.) in full HD resolution (1920 ×1080), filmed on a Panasonic HC-V730 video camera, 24.0 megapixels deep. The accuracy of determining breakdowns of elements increased by 20%; more detailed information is given in Table 2 for each object.

Within the development of the smart monitoring system, the DDT of the main equipment were developed; some models were developed manually and some in automatic mode. All models can be kept up to date in automatic mode. Figure 5 shows the user interface of the intelligent video surveillance system. The information comes from three sources (sensors, video cameras and microphones), escalator breakdowns are displayed in the center of the application, and the current state of the escalator is shown on the right. On the left are the log lines, and at the bottom are the process control buttons (start, stop and restart).

After several months of experimental operation, statistics were obtained.

An analysis of the statistics showed that the recovery time of the escalator equipment decreased by an average of 20%.

The results of the comparison are shown in Table 2. The average recovery time was defined as the time of delay between the occurrence of a malfunction and the time of making a decision on the required actions. Previously, these decisions were made by the dispatchers. After the deployment of the system based on the use of DDT, reconfiguration decisions can be made manually, semi-automatically or in automatic mode. 

The use of DDT helped solve the problems associated with the diversity of equipment.

At the next phase, it is planned to use DDT to solve the following tasks: (i) creation of a system for staff training; (ii) accumulation of statistics on malfunctions; and (iii) creation of a system for assessing the “health of the system”.

## 11. Conclusions

This article proposes an approach for the construction of DDTh. A distinctive feature of the proposed approach is the use of the DDT paradigm developed by the authors [18], which is based on the use of structural and functional models.

The idea of the proposed approach is the fusion of three known paradigms: DT, EA and AA. The use of DT must enable a reduction in the total cost of ownership. EA is considered a mechanism for adapting the system to changes in the external environment. AA is considered as a means of maintaining RT agility. 

The specific features of the proposed approach are the following: (i) DT are considered as classes, whereas DS are considered as instances of classes; (ii) as a formal model describing the process of architectural design of DDTh, it is proposed to use a three-level model that includes the following levels: DDTh support level, AA support level, and DS support level. 

The proposed DDTh approach was used for developing real systems, including in the production system of an IIoT platform [21], the monitoring and management system for cable television [22,25], and an intelligent video surveillance system for the subway. Currently, an educational content management system, based on the proposed approach, is being developed. In all cases, the proposed approach has shown its efficiency and effectiveness. 

The proposed approach may be of interest to specialists engaged in research and development in the field of IoT and IIoT, primarily architects. 

The experience gained in the process of real system development shows that the suggested DDTh approach can be used to support the DDTh paradigm at the stages of exploitation and modernization.

Currently, work on further development of the DDTh approach is being carried out in four directions: (i) development of algorithms for the transformation of models, and the construction of domain-oriented models and model patterns; (ii) creation of libraries of model patterns; (iii) expansion of the scope of the DDTh approach, in particular, for the analysis of social networks; (iv) the use of models developed at the design stage during the operation and modernization of DDTS, in accordance with the ideas of DevOps [12].

## Figures and Tables

**Figure 1 sensors-22-03078-f001:**
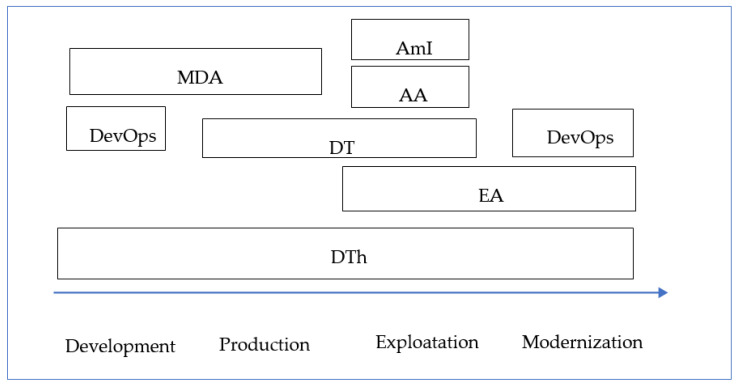
The use of the paradigms related to different stages of the LC.

**Figure 2 sensors-22-03078-f002:**
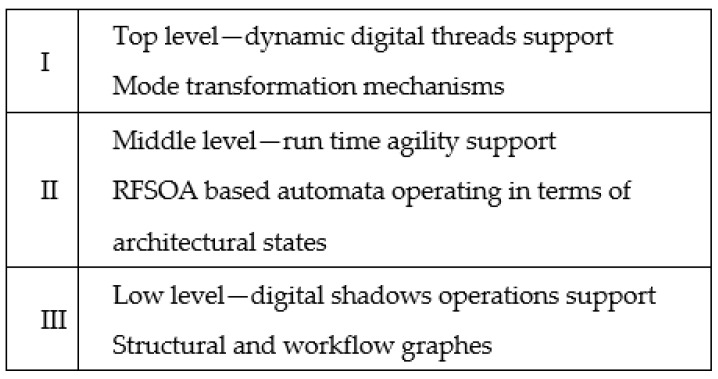
Three-level DDTh model.

**Figure 3 sensors-22-03078-f003:**
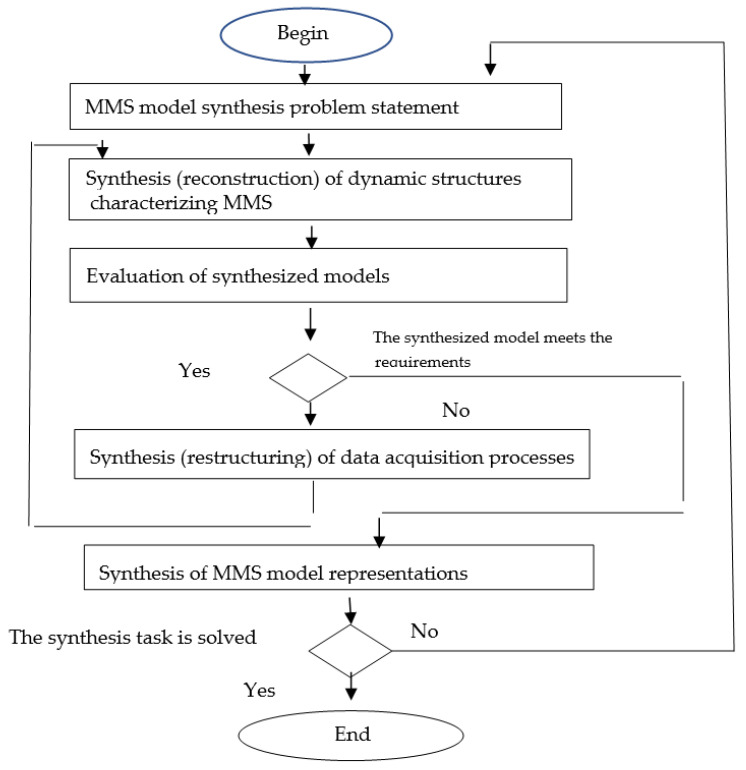
Generalized algorithm of multilevel synthesis of MMS models.

**Figure 4 sensors-22-03078-f004:**
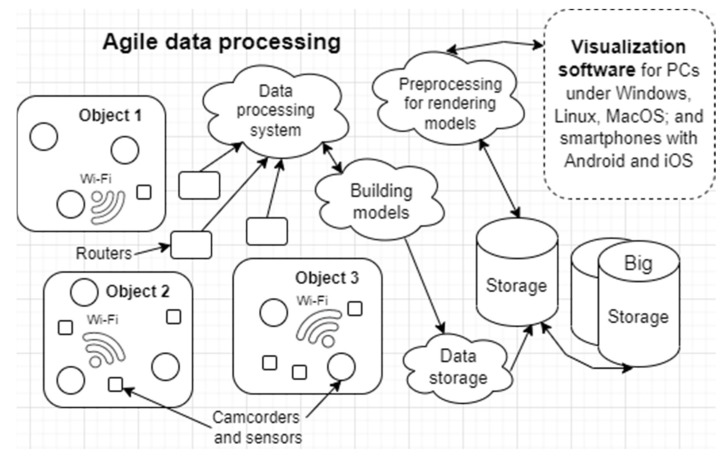
The structure of the intelligent video surveillance system.

**Figure 5 sensors-22-03078-f005:**
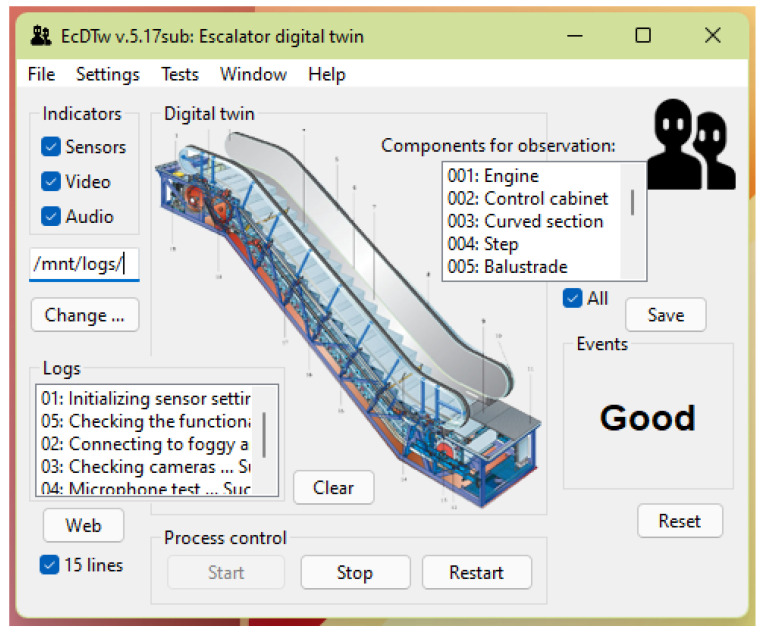
Escalator DT.

**Table 1 sensors-22-03078-t001:** Main types of models.

N	Model Type
1	V→V (VV mapping)
2	V→P (VP mapping)
3	P→P (PP mapping)
4	V→B (VB mapping)
5	V→S (VS mapping)

**Table 2 sensors-22-03078-t002:** Time required to repair breakdowns.

№	Element Name	Number of Incidents	Average Recovery Time, s		Gain %
			Without DT	With DT	
1	Engine	4	197	176	10.65
2	Control cabinet	12	246	210	14.63
3	Curved section	23	179	157	12.29
4	Step	59	163	135	17.17
5	Balustrade	47	42	33	21.42
6	Backlight apron	72	54	42	22.22
7	Apron	54	381	308	19.16
8	Handrail	82	692	610	11.84
9	Entrance area	91	284	233	17.95
10	Control buttons	67	126	94	25.39
11	Floor slab	58	782	610	21.99
12	Stretching device	42	935	684	26.84
13	Handrail mouth	31	629	487	22.57
14	Including thrust of step interlocks	17	538	355	34.01
15	Step roller guides	13	491	404	17.71
16	Escalator truss	6	942	738	21.65
17	Chain of steps	38	821	636	22.53
18	Handrail drive	27	782	613	21.61
19	Escalator drive	13	629	492	21.78
Average					20.1794

## Data Availability

https://github.com/alex1543/Twins (accessed on 5 April 2022).

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
