# Peer review of "Towards Dynamic Model-Based Agile Architecting of Cyber-Physical Systems"

_sensors, 2022, doi:10.3390/s22083078_

Round 1

Reviewer 1 Report

  • I strongly recommend adding the chapter arrangement to the introduction.
  • In the introduction, I cannot see the urgent need for this research, for example, reliability is critical for IoTs but not mentioned. Therefore, to refine the background, some existing work can help: “Modeling and analyzing cascading failures for Internet of Things”, “Toward robust and energy-efficient clustering wireless sensor networks: A double-stage scale-free topology evolution model”, “Cascade Failures Analysis of Internet of Things under Global/Local Routing Mode”.
  • Please double check the writing conventions in this paper, for example, on page 3, “object» ” should be “ object” ”?
  • Figure 2 should receive a more complete modification. For example, what devices are the top-level composed of, what functions it has and what tasks it needs to accomplish?

Author Response

Author's Reply to the Review Report (Reviewer 1)

Item

Reviewer’s comments

Reply

1

I strongly recommend adding the chapter arrangement to the introduction.

Corrected.

The description of the article content is added.

2

Please double check the writing conventions in this paper, for example, on page 3, “object» ” should be “ object” ”?

Corrected

3

In the introduction, I cannot see the urgent need for this research, for example, reliability is critical for IoTs but not mentioned. Therefore, to refine the background, some existing work can help: “Modeling and analyzing cascading failures for Internet of Things”, “Toward robust and energy-efficient clustering wireless sensor networks: A double-stage scale-free topology evolution model”, “Cascade Failures Analysis of Internet of Things under Global/Local Routing Mode”.

First of all, it should be noted that this comment includes two separate sub comments: i) the relevance of research topics is not obvious, and ii) the problem of cascading failures is not considered.

The relevance of the proposed approach is based on the fact that a distinctive feature of modern anthropogenic systems, along with a complex structure and behavior, is the high level of structural and functional dynamics, i.e. the structure and behavior of the systems are permanently changing in time. This makes it difficult to solve many tasks, such as data acquisition and management at different levels, especially when we work with system of systems. The problem of reliability in the framework of the proposed approach is considered indirectly as follows. The most important sub characteristic of the reliability is availability. It can be increased through the use of mechanisms of both structural and functional reconfigurations. The proposed approach is focused specifically on working with systems with high structural and functional dynamics.

Changes have been made in the text in order to make this point more understandable.

The problem of cascading failures is a fairly well-known problem. Within the framework of this approach, it is not directly considered. It is assumed that this task is solved at the stage of preprocessing information about events received in the form of logs. Different approaches can be used to process the event streams. In our projects we use both rule based systems and neural networks. But this problem is not in the scope of this article.

The proper information has been added to the text of the article and a reference has been added.

4

Figure 2 should receive a more complete modification. For example, what devices are the top-level composed of, what functions it has and what tasks it needs to accomplish?

Figure 2 is corrected.

Reviewer 2 Report

The article under review deals with the interesting and topical subject of discussing the architectural appearance of future SoS based on the Digital Threads concept. The paper corresponds to the journal's topic, written in a good language, with illustrations and a good list of references. A significant drawback of the article is the lack of a dedicated "methodology" chapter, although part of the information about methods and approaches is presented in the introduction. There are some comments to which the authors should pay attention.  

Some possible minor issues:

295: OMS models and its 295 elements are considered as first-class citizens Is it correct term in this context - "first-class citizens"? Please confirm 

298 and later: DDTh and DDth - if it is the same, better to use single form of acronym.

The article can be published in the journal after possible clarification

Author Response

Author's Reply to the Review Report (Reviewer 2)

Item

Reviewer’s comments

Reply

1

A significant drawback of the article is the lack of a dedicated "methodology" chapter, although part of the information about methods and approaches is presented in the introduction. There are some comments to which the authors should pay attention. 

Corrected. The proper text is added to the Section 5.

2

295: OMS models and its 295 elements are considered as first-class citizens Is it correct term in this context - "first-class citizens"? Please confirm

Unsuccessful expression.

Corrected.

3

298 and later: DDTh and DDth - if it is the same, better to use single form of acronym.

Corrected. Correct abbreviation is DDTh

4

Figure 3 and Figure 4 and Figure 5 have got rare connectivity, and so are the conclusions

These figure are linked in a following way. Figure 3 presents the structural synthesis algorithms. It is one of the basic algorithms. Figure 4 presents the structure of intelligent video surveillance system in the frames of which this algorithm is used.

Figure 5 shows Escalator DT user interface.

The proper text is added.

5

No results and moreover, one loses connectivity in reading while reading the paper and have rarely heard terminologies such Observed and Managed Systems (OMS) or Digital Shadow (DS).

Authors agree with this remark and have done all the best to improve the situation. The abbreviation Monitored and Managed System (MMS) is used. The term Digital Shadow (DS) was used in old times as a synonym of the term Digital Twin. We use this term DS for defining the observed object current state (snapshot). We need this term but we do not know the commonly used term for defining this concept.

Reviewer 3 Report

  • The paper presents microelectronics, telecommunications and software engineering as the main driving force of current stage of development in engineering technology.
  • Describing the increase in number of element, internal and external connections and continuous increase in the level of variability as one reason of man-made complexity of system
  • Another reason of man-made complexity is that these system are made from different physical nature such cyber-physical system, socio-cybernetic systems and natural system
  • Many man-made features are implementing complex set of functions through software, using many sensor and calculators which lead to accumulate and process large amounts of data, referred to as big-data problems.
  • So one can regard modern complex systems are due to high level of structural and functional variability as well as high level of heterogeneity.
  • Modern system can be right called as Large Scale Complex Distributed System (LSCDS) and can be considered as a System of Systems are intelligent too and hence require approaches and paradigms.
  • The paper considers one of the possible approaches to the development of distributed systems built on IoT platforms with Research Objectives, namely, RO1 is development of approaches to the implementation of the mechanisms of the Digital Threads (DTh) and RO2 is development of mechanisms to support Run Time (RT) agility.
  • Complexity and variability of LSCDS systems could be Model Driven Architecture (MDA) [9], EA [8], agile architecture (AA) [10,11], DevOps [12], DT 13], DTh [14,15], Ambient Intelligence (AmI) [4].
  • The idea of the proposed approach is the joint use of three paradigms: DTh, EA and AA. EA is considered as a means of adapting the system to changes in the external environment. AA is considered as an instrument of maintaining RT agility. The proposed approach can be defined as the Agile Digital Threads (ADTh) approach.
  • This project is a part of a big project "Smart City", one of the goals of which is the management of transport and passenger flows in St. Petersburg (Russia), describing
  • The paper appears to be too much conceptual, and the authors have by their another paper

Vodyaho, Alexander Ivanovich, Nataly Alexandrovna Zhukova, Yulia Alexandrovna Shichkina, Fahem Anaam, and Saddam Abbas. "About One Approach to Using Dynamic Models to Build Digital Twins." Designs 6, no. 2 (2022): 25.

  • No results and moreover, one loses connectivity in reading while reading the paper and have rarely heard terminologies such Observed and Managed Systems (OMS) or Digital Shadow (DS).
  • Figure 3 and Figure 4 and Figure 5 have got rare connectivity, and so are the conclusions
  • The Abstract says

This approach has been used for development of a number of real systems and 16 has shown its effectiveness. The described approach is focused on the use of large-scale distributed systems for various purposes built on IoT and IIoT platforms and may be of interest to specialists 18 engaged in research and development in the domain of IoT and IIoT based information systems, 19 primarily for architects – one would to read some case studies accordingly.

  • The Conclusion writes

The idea of the proposed approach includes three key points: I) OMS models and its 295 elements are considered as first-class citizens, while DT are considered as classes, DS are 296 considered as instances of classes; II) as a formal model describing the process of architec-297 tural design of DDth it is proposed to use a three-level model that includes the following 298 levels: I) DDTh support level, II) AA support level, III) DS support level.

Author Response

Author's Reply to the Review Report (Reviewer 3)

Item

Reviewer’s comments

Reply

1

Figure 3 and Figure 4 and Figure 5 have got rare connectivity, and so are the conclusions

These figures are linked in a following way. Figure 3 presents the structural synthesis algorithms. It is one of the basic algorithms. Figure 4 presents the structure of intelligent video surveillance system in the frames of which this algorithm is used.

Figure 5 shows Escalator DT user interface.

2

The paper appears to be too much conceptual, and the authors have by their another paper

Vodyaho, Alexander Ivanovich, Nataly Alexandrovna Zhukova, Yulia Alexandrovna Shichkina, Fahem Anaam, and Saddam Abbas. "About One Approach to Using Dynamic Models to Build Digital Twins." Designs 6, no. 2 (2022): 25.

The presented article is really based on the mentioned paper where only very general idea of the digital twin approach is described.

The presented paper is devoted to the digital threads that can be considered as a development of digital twin approach. The reference is added.

3

There is mismatch between annotation, introduction and conclutions

Annotation and conclusions are corrected.

Round 2

Reviewer 3 Report

Is the subject matter presented in a comprehensive manner?

I am not convinced if the authors have done justice to support title “Towards Dynamic Model-Based Agile Architecting of Cyber-Physical Systems,” which is better that its previous version “Towards Agile Digital Threads Architecting”. Readers do not get convinced if the paper has presented a meaningful outcome, the conclusion is not reflective of what is promised in the Abstract. I suggest the authors must include some case studies examples which may become a good paper for future submission – it is too abstract kind of an idea but not having a physical or concrete existence that has yet to be proven as a physical reality. The Authors further write in conclusion that they are working on an educational content management system based on the proposed approach is being developed. In all cases, the proposed approach has shown its efficiency and effectiveness, and let us hope that would be their some system-based implementation of this approach.  

Are the references provided applicable and sufficient?

The authors take support in the revised version from twenty nine (29) instead of thirty two (32) in the previous version current and branded references, including one (1) from MDPI.

Further, the Conclusion does not seem to be drawn based on the Results and Discussion, making sure that what has been promised in Abstract is delivered in Conclusion.” I do not see the authors have done this job effectively, though the current Towards Dynamic Model-Based Agile Architecting of Cyber-Physical Systems” compared to its previous version “Towards Agile Digital Threads Architecting”.}

Author Response

The Reviewer 3 asks to explain the novelty of the suggested article "Towards Agile Digital Threads Architecting" in comparison with the article "About One Approach to Using Dynamic Models to Build Digital Twins" published by the authors. We are agree that both articles discuss the similar model based approach. The second article ("Towards Agile Digital Threads Architecting") is based on the first article (About One Approach to Using Dynamic Models to Build Digital Twins). The first article discusses one of the possible approaches to the run time digital twins realization, i.e. it describes one of the possible approaches to the cyber-physical systems with agile architecture implementation. The description of this approach one can find e.g. in [Bloomberg, J. The Agile Architecture Revolution: How Cloud Computing, REST-Based SOA, and Mobile Computing Are Changing Enterprise IT; Wiley & Sons, Inc.: Hoboken, NJ, USA, 2013; 278p] , but in this book a realization is not described. In this case our model based approach is conceded as an instrument of integration of elements of different physical nature. In the second article (Towards Agile Digital Threads Architecting) the problem of digital threads building is discussed, i.e. the problem of model set building. It is a problem of systems continuous arcitecturing [Erder, M.; Pureur, P.; Woods, E. Continuous Architecture in Practice Software Architecture in the Age of Agility and DevOps; Upper Saddle River, NJ. USA, 2021; 321 p.].

To our mind it is different problems.

In order to clarify this moment a proper corrections in the article text are made.

Round 3

Reviewer 3 Report

  • The paper presents microelectronics, telecommunications and software engineering as the main driving force of current stage of development in engineering technology.
  • Describing the increase in number of element, internal and external connections and continuous increase in the level of variability as one reason of man-made complexity of system
  • Another reason of man-made complexity is that these system are made from different physical nature such cyber-physical system, socio-cybernetic systems and natural system
  • Many man-made features are implementing complex set of functions through software, using many sensor and calculators which lead to accumulate and process large amounts of data, referred to as big-data problems.
  • So one can regard modern complex systems are due to high level of structural and functional variability as well as high level of heterogeneity.
  • Modern system can be right called as Large Scale Complex Distributed System (LSCDS) and can be considered as a System of Systems are intelligent too and hence require approaches and paradigms.
  • The paper considers one of the possible approaches to the development of distributed systems built on IoT platforms with Research Objectives, namely, RO1 is development of approaches to the implementation of the mechanisms of the Digital Threads (DTh) and RO2 is development of mechanisms to support Run Time (RT) agility.
  • Complexity and variability of LSCDS systems could be Model Driven Architecture (MDA) [9], EA [8], agile architecture (AA) [10,11], DevOps [12], DT 13], DTh [14,15], Ambient Intelligence (AmI) [4].
  • The idea of the proposed approach is the joint use of three paradigms: DTh, EA and AA. EA is considered as a means of adapting the system to changes in the external environment. AA is considered as an instrument of maintaining RT agility. The proposed approach can be defined as the Agile Digital Threads (ADTh) approach.
  • This project is a part of a big project "Smart City", one of the goals of which is the management of transport and passenger flows in St. Petersburg (Russia), describing
  • The paper appears to be too much conceptual, and this papers seems to be an extension of authors’ previous paper listed as udner:

Vodyaho, Alexander Ivanovich, Nataly Alexandrovna Zhukova, Yulia Alexandrovna Shichkina, Fahem Anaam, and Saddam Abbas. "About One Approach to Using Dynamic Models to Build Digital Twins." Designs 6, no. 2 (2022): 25.

  • Thought there are no results and moreover, but one finds that the rarely heard terminologies such Observed and Managed Systems (OMS) or Digital Shadow (DS), could be leading to systems development of the distant future.
  • Figure 3 and Figure 4 and Figure 5 have got rare connectivity, and but the conclusions and Abstracts are well-unconnected
  • The Abstract says

This approach has been used for development of a number of real systems and has shown its effectiveness for supporting system agility on the stages of exploitation and modernization. The proposed approach is focused on building digital twins based systems. The article can be 19 interesting for specialists engaged in research and development in the domain of IoT and IIoT based information systems, primarily for architects.

  • The Conclusion writes

The proposed approach may be of interest to specialists engaged in research and development in the field of IoT and IIoT, primarily for architects, and further it says that the experience received in the process of real systems development shows that the 342 suggested DDTh approach can be used to support DDTh paradigm on the stages of exploitation and modernization.

The authors have presented concepts and terminology in the paper titled “Towards Dynamic Model-Based Agile Architecting of Cyber-Physical Systems” compared to its previous version “Towards Agile Digital Threads Architecting”. The paper may be helpful for some system-based implementation by some potential system developers, and may be recommended for acceptance.
